# A diamond-bearing core-mantle boundary on Mercury

Yongjiang Xu [1], Yanhao Lin [1] ✉, Peiyan Wu[1,2], Olivier Namur[3], Yishen Zhang[3] & Bernard Charlier [4]

Abundant carbon was identified on Mercury by MESSENGER, which is interpreted as the remnant of a primordial graphite flotation crust, suggesting that the magma ocean and core were saturated in carbon. We re-evaluate carbon speciation in Mercury's interior in light of the high pressure-temperature experiments, thermodynamic models and the most recent geophysical models of the internal structure of the planet. Although a sulfur-free melt would have been in the stability field of graphite, sulfur dissolution in the melt under the unique reduced conditions depressed the sulfur-rich liquidus to temperatures spanning the graphite-diamond transition. Here we show it is possible, though statistically unlikely, that diamond was stable in the magma ocean. However, the formation of a solid inner core caused diamond to crystallize from the cooling molten core and formation of a diamond layer becoming thicker with time.

Spectral data from the MESSENGER spacecraft revealed that the unusual darkness of Mercury's surface is due to the widespread presence of graphite[1]. Measurements by neutron and gamma-ray spectroscopy quantified the abundance of carbon in the crust to ca. 2–4 wt%[2,3], although a recent reinvestigation of the data suggests lower concentrations (<1 wt.%)[4]. The close association of graphite with lower crustal material observed in deep craters supports an endogenous origin for graphite[5,6] and rules out major exogenous contributions by comets[7].

Because of the highly reduced conditions that prevailed during Mercury's differentiation (in the range IW − 2.6 to IW − 7.3[8,9], reported in log units relative to iron-wüstite thermodynamic equilibrium, IW), light elements in Mercury's iron core are thought to be dominated by silicon, sulfur, and carbon[10–13]. Although carbon likely degassed from the magma ocean as $CO_2$, CO, and $CH_4$ species[14–18] and was lost to space, the abundance of graphite in the Mercurian crust indicates that the planet remained saturated in a carbon phase during metal-silicate differentiation, core formation, and the entirety of magma ocean crystallization[6]. Additionally, because the solubility of carbon in silicate melt is exceptionally low under reduced conditions[19,20], a significant amount of excess carbon should have been available to contribute to the formation of a graphite flotation crust.

It has so far been assumed that the pressure-temperature profile of Mercury's mantle and magma ocean did not enter the diamond stability field due to the lower pressure at Mercury's core-mantle boundary (CMB) and the absence of accurate constraints on the magma ocean liquidus temperature. As such it was considered that graphite was the only stable carbon-bearing phase during magma ocean crystallization[21]. Because graphite ($\rho \approx 2200$ kg m$^{-3}$) is less dense than a magma ocean melt ($\rho \approx 2700$ kg m$^{-3}$), it is expected to have floated and contributed to the formation of Mercury's primordial crust[13,20], analogous to the formation of the primordial anorthosite crust on the Moon[22,23]. However, the nature of the C-bearing phase requires reassessment in light of the most recent gravity field models for Mercury[24–26]. A smaller normalized polar moment of inertia (MOI) of the entire planet has been proposed ($0.333 \pm 0.005$[24]). Although this latter value is close to the −1$\sigma$ error value of the classical estimate ($0.346 \pm 0.014$[27]), a model with a lower MOI would provide a deeper core-mantle boundary (CMB), and thus assumedly a deeper interface between the core and the magma ocean: $485 \pm 20$ km[25] (similar to values of Steinbrügge et al.[28]) compared to $436 \pm 25$ km[26] when using a higher MOI, similar to other studies[29,30] (Mercury's mean radius being 2440 km[31]). Such a change in the depth of the CMB will influence the

[1]Center for High Pressure Science and Technology Advanced Research, Beijing 100193, People's Republic of China. [2]School of Earth Sciences and Resources, China University of Geosciences, Beijing 100083, People's Republic of China. [3]Earth and Environmental Sciences, KU Leuven, 3001 Leuven, Belgium. [4]Department of Geology, University of Liege, Sart Tilman, Liege 4000, Belgium. ✉e-mail: yanhao.lin@hpstar.ac.cn

pressure at the CMB that remained to be quantified and thus an effect on the stability of carbon phases.

Here, we propose a new thermodynamic estimate of Mercury's magma ocean temperature accounting for the depression of the silicate melt liquidus in the presence of significant amounts of sulfur. Based on the revised geodetic calculations, we also recalculate the temperature and pressure at Mercury's CMB. Informed by experiments at the relevant conditions and our new thermodynamic models, we revise the relative stability of graphite and diamond at the present CMB and in the magma ocean. We also evaluate the role of core crystallization and the implications for the exsolution of a carbon-bearing phase from the molten outer core. The unique reduced conditions of Mercury, the saturation of carbon phase in different reservoirs, and the formation of a solid inner core allow for different scenarios capable of producing a diamond layer at Mercury's CMB.

## Results & Discussion

### Pressure at the CMB and the liquidus of the deep magma ocean

We used planetary interior structure models that satisfy the various measurements of MOI[25] to calculate that the pressure at Mercury's CMB is 5.77 ± 0.31 GPa with a low MOI[24] and 5.38 ± 0.37 GPa with a high MOI[26] (Methods and Supplementary Fig. S1). According to our calculations, the highest possible pressure of Mercury's CMB is 7 GPa. We here provide experimental melting relations at 7 GPa to determine experimentally the carbon speciation at the most extreme pressure conditions of Mercury's magma ocean. Experiments are then combined with thermodynamic modelling to investigate how phase relations change for lower pressure CMB conditions which would still match Mercury's MOI.

The most relevant estimate for the composition of the primordial Mercurian mantle is the silicate fraction of enstatite chondrites (EH-EL chondrites[32,33]). Thus, we compiled the chemical compositions of EH-EL meteorites[34,35], excluding iron metal and decreasing phosphorous concentration (to 0.1 wt% $P_2O_5$) relative to the bulk EH-EL contents because of its siderophile behavior at reduced conditions[36]. Because silicon is known to partition more readily into metallic melts under highly reduced conditions[11,37], we selected two starting compositions representing the partitioning of 8 and 15 wt% Si into the metallic core (Mer8 and Mer15, respectively; Supplementary Table S1). These values span the range of the most consistent interior structure and thermal models for Mercury[25,38]. Because sulfur is present in significant amounts at the surface of Mercury[39,40] and has lithophile behavior under reduced conditions[10,41], we considered sulfur as a potential major element in the Mercurian magma ocean (and a minor element in Mercury's core) and added FeS to the starting materials to ensure sulfide saturation in the silicate melts. Experiments were performed at 7 GPa using a cubic multi-anvil press to understand melting relations at the greatest possible depths within the magma ocean (Methods and Supplementary Fig. S2). This pressure corresponds to the maximum value of potential CMB pressures predicted by interior models[25]. The liquidi of the Mer8 and Mer15 compositions at 7 GPa were 2188 ± 15 K and 2213 ± 10 K, respectively (Fig. 1a). Mer8 was in the stability field of orthopyroxene whereas Mer15 first crystallized olivine. With cooling, both compositions reached a cotectic surface with olivine + orthopyroxene, followed by garnet and clinopyroxene. A sulfide phase was present in all experiments (FeS + MgCaFeS). The solidi of Mer8 and Mer15 at 7 GPa were 2023 ± 50 K and 2113 ± 35 K, respectively. The calculated oxygen fugacities in our experiments were IW − 3.9 to IW − 5.1 (Methods, Supplementary Fig. S3, and Supplementary Data 1).

### The role of sulfur on liquidus depression

Using the MAGEMin thermodynamic calculator[42] implementing the most recent THERMOCALC thermodynamic model[43], and assuming that the pressure at the bottom of Mercury's magma ocean was similar to the present CMB pressure of 5.77 ± 0.41 GPa, we calculated the silicate melt liquidi across the pressure range relevant to Mercury's magma ocean. Thermodynamic models have previously been shown to be very accurate to reproduce phase equilibria of sulfur-free Mercury-like magmas[44]. We used the composition of a magma ocean in equilibrium with a core containing 8 wt% Si. We then used our experimental dataset to parametrize the effect of sulfur on the liquidus (Methods, Supplementary Fig. S5, S6). The incorporation of 1 wt% S into the silicate melt under the reduced conditions of Mercury depresses the liquidus by 59 K, and the liquidus is depressed further, but by a decreasing degree, with increasing S content (Fig. 1b). Our experimental liquids contained 3.08–16.23 wt% S and the Mercurian magma ocean is predicted to contain up to 11 wt% S[10], depressing the liquidus by up to 358 K compared to the S-free system (Fig. 1b). It has been shown previously that sulfur solubility in reduced silicate melts is not pressure sensitive[10] so that the liquidus depression we calculate as a function of sulfur content is independent on pressure conditions. Although iron is partitioned into the core under the reduced conditions of Mercury, making the magma ocean Mg-rich, the major effect on the liquidus is the incorporation of sulfur as a major element. Carbon solubility in the reduced magma ocean is orders of magnitude lower than that of sulfur[10,19] which implies that carbon has no effect on the magma ocean liquidus temperature.

### Diamond in Mercury's magma ocean

Based on the pressure-temperature conditions at the onset of crystallization in a carbon-saturated magma ocean, we calculated whether graphite or diamond was likely to be stable using the thermodynamic model of Day[45]. Temperature was obtained using the equation for the liquidus with 7 and 11 wt% S, pressure was calculated for each interior structure model (Methods). For a S-free magma ocean, all CMB pressure estimates plot in the graphite stability field. In contrast, if the silicate melt contained 7 or 11 wt% S, 0.6% or 8.9% of the pressure estimates, respectively, plot in the diamond stability field. We also tested the probability that diamond was stable at 50% crystallization if solidification of the magma ocean was governed by equilibrium crystallization[46–48]. Under those conditions, 2.5% and 20.6% of the models produce diamond at 7 and 11 wt% S, respectively. Therefore, most models support graphite precipitation during magma ocean solidification and thus a primordial graphite flotation crust[21]. Although not impossible, the production of diamond from the silicate magma ocean and its sinking to the CMB is statistically improbable. Carbon solubility in reduced magma ocean (<IW-3) is low when considering CO and $CO_2$ solubility (< 3 ppm; Supplementary Fig. S7). Recent experiments have shown that it could be slightly higher (up to 15 ppm) under very reduced conditions compared to thermodynamic models[19]. This may be due to a minor $CH_4$ species in the melt. Nevertheless, the low solubility of carbon under reduced conditions[19,20] means that any amount of diamond produced during the early stages of magma ocean crystallization would have been minor: we calculated that diamond crystallization in the magma ocean within 1 GPa above the CMB at IW − 7 to IW − 3 could only have produced a diamond layer 0.1 to 200 m thick, respectively and depending on the considered C solubility model (Methods, Supplementary Fig. S8). We also constrained the thickness of the graphite layer that could have been produced by carbon saturation during magma ocean crystallization to 2 to 2000 m, also depending on oxygen fugacity and the considered C solubility model (Methods, Supplementary Fig. S7). However, carbon solubility in the magma ocean does fix the lower limit for the thickness of the primordial graphite crust as graphite delivered to Mercury by the building blocks may have accumulated at the surface of the molten planet at carbon-saturation[6].

### Diamond from the core

Excess graphite present at the surface of Mercury suggests that the entire planet, including its core, was saturated in carbon during its primordial stage. The carbon concentration in the core of a C-saturated Mercury can

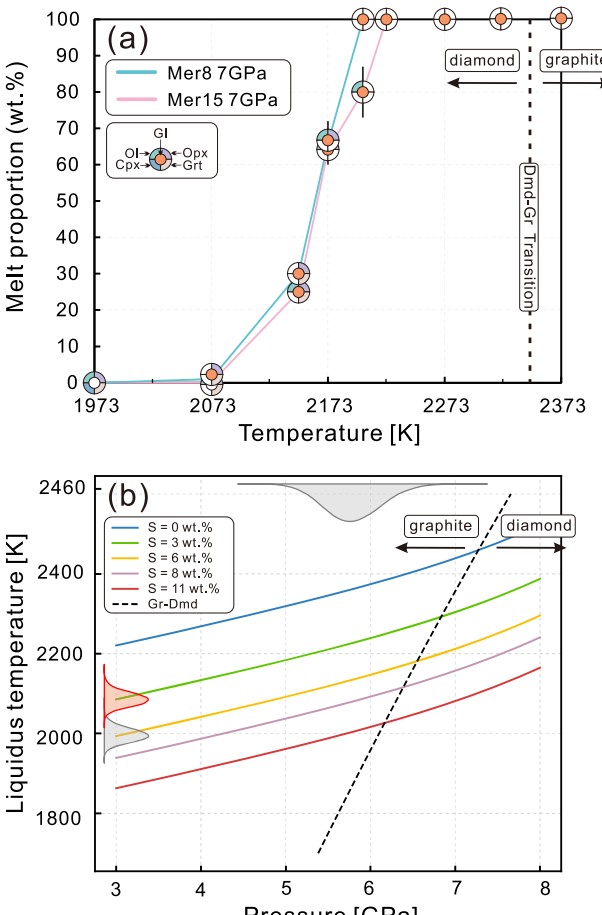

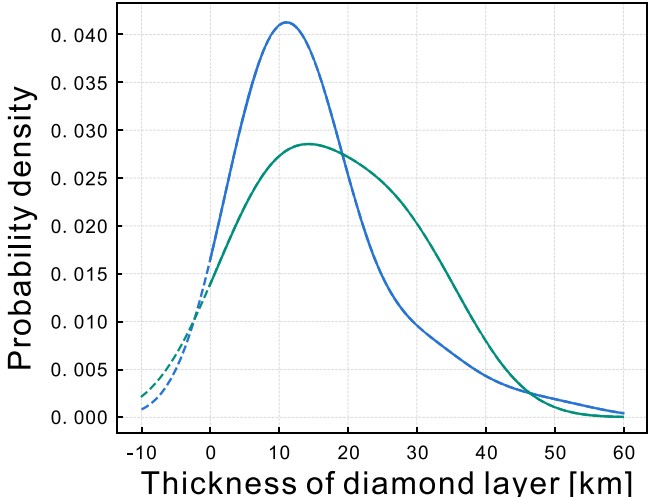

**Fig. 2 | Thickness of the diamond layer at the core-mantle boundary.** Calculated using interior models[25] with MOI = 0.333 ± 0.005[24] (blue line) and 0.343 ± 0.006[26] (green line). The mean calculated thickness of the diamond layer at CMB today is -14.9–18.3 km on average with an uncertainty of 10.6 km. The model considers a solid FeSi core with 2.1 wt% carbon.

**Fig. 1 | Experimental results and the impact of sulfur on depressing the liquidus temperature. a** Melting and phase relations for compositions Mer8 and Mer15 as a function of temperature at 7 GPa; solid lines highlight the evolving melt fraction with decreasing temperature. The vertical dashed line at 2356 K is the diamond-graphite transition at 7 GPa[45], consistent with Raman spectra of 'graphite' capsules in our experiments at 2323 and 2373 K (Supplementary Fig. S4), demonstrating that the experimental pressure is reliable. Abbreviations for phase assemblages: Dmd diamond; Gr graphite; Gl glass; Ol olivine; Opx orthopyroxene; Cpx clinopyroxene; Grt garnet. **b** Model illustrating the impact of sulfur in depressing the liquidus temperature in Mercury's magma ocean. The probability density distribution along the x-axis (top) indicates the range of pressures calculated at the present CMB using the interior models of Goossens et al.[25], and those along the y-axis indicate the range of temperatures calculated at the base of a magma ocean containing 7 and 11 wt% S at pressures calculated from interior models[25] (red and gray, respectively).

be constrained by the anticorrelation between the abundances of silicon and carbon in the iron-rich metallic core [e.g., refs. 13,14]. Using a compilation of experiments on metal-silicate equilibration (Methods, Supplementary Fig. S9), we calculated a probability density distribution for carbon concentrations in the core based on the silicon concentration in the core[25] and obtained an average value of 3.4 ± 1.0 wt% C using MOI = 0.333 ± 0.005[24] and 1.67 ± 1.0 wt% C using MOI = 0.343 ± 0.006[26] (Methods, Supplementary Fig. S10). Geodetic measurements support the existence of an inner solid core with a radius 30–70% that of the outer core[24,29,49]. Experimental studies have shown that the inner core should be a FeSi phase[50–52]. The formation of the solid inner core would have caused diamond to crystallize from the residual metallic molten core. This is because crystallization of C-poor phases[52] in the solid core would have for effect to enrich the liquid outer core in C. However, the whole core being at C-saturation implies that the C-enrichment will be accommodated by forming diamond in thermodynamic equilibrium with the liquid core

which would then have floated to the CMB due to the much lower density of diamond compared to the liquid iron (+Si, +C, +S) alloy. Using the size of the inner core from interior model[25] and considering it to be made of FeSi solid phase with 2.1 wt% C[52] and the total carbon content of the fully molten core (Methods), we calculate a CMB diamond layer thickness of 14.9 ± 10.6 km using a MOI = 0.333 ± 0.005[24] and 18.3 ± 10.6 km with MOI = 0.343 ± 0.006[26] (Fig. 2). Carbides (Fe₃C and Fe₇C₃) have also been proposed as potential products of core crystallization on Earth[53,54]. Because the crystallization regime in the molten core of Mercury involves the formation of Fe(Si) snow[55,56], the shallow pressure and temperature conditions at CMB must be considered when evaluating the nature of crystallization products. Fe₃C may be stable below 7 GPa and is replaced by Fe₇C₃ at higher pressures[54–59]. However, it has been shown that Fe₃C melts in a peritectic reaction to form liquid + diamond at 1688 K at 5.7 GPa[60]. This is further supported by other studies which shows that at 5 GPa, Fe₃C melts at ca. 1650 K[57,58]. All these studies support that, as compared to Earth, low-pressure core conditions on Mercury do not favor the formation of carbides which would be stable at lower temperatures.

Any diamond formed at the inner core interface and floated to the CMB very early on in the cooling history of Mercury might have transformed into graphite if CMB temperature was too high. However, compared to the primordial stage of magma ocean crystallization, the CMB today is colder[61]. Consequently, although the temperature at the CMB 4.5 Ga was at the upper limit for diamond stability, the CMB may now be within the diamond stability field due to secular cooling, even at the lowest estimated CMB pressure. Indeed, the probability distribution of present-day CMB temperatures[25], which is consistent with independent estimates based on thermal evolution models[61], nearly matches the graphite-diamond transition at the pressure of the CMB (Methods, Supplementary Fig. S11). We thus propose that the CMB today may be at the graphite-diamond transition, which may buffer the CMB temperature. The potential diamond layer thicknesses shown in Fig. 2 are thus maximum values because the extent of the graphite to diamond reaction cannot be predicted and because some early formed graphite may have been redistributed in the mantle due to strong convection and density contrast between graphite and silicate mantle minerals. In addition, convection in the lower mantle may have disrupted the diamond layer and redistributed diamond in the mantle or even in the crust. However, we believe that graphite and diamond redistribution in the silicate part of the planet was likely minimal. This is because strong convection in the

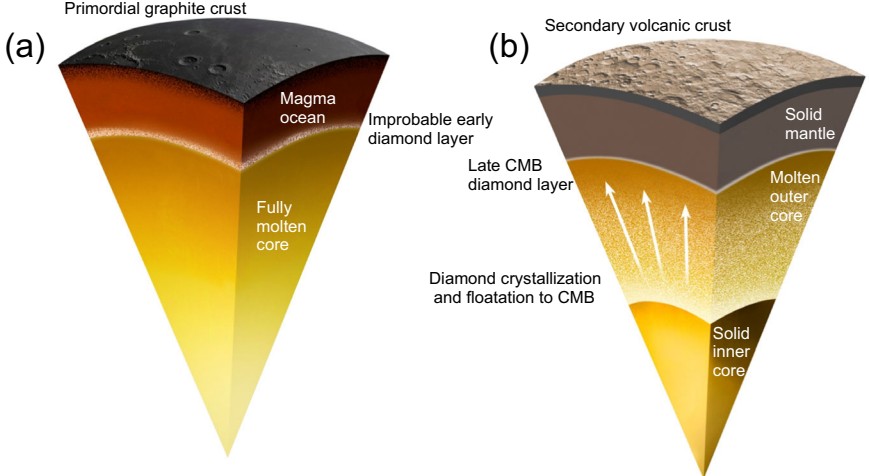

**Fig. 3 | Proposed scenario for the formation of diamond at Mercury's core-mantle boundary. a** Crystallization of the carbon-saturated silicate magma ocean and the potential, yet unlikely, early production of diamond at its base. Graphite was the major phase to form in the magma ocean and accumulated at the surface to form a primordial graphite crust. **b** During crystallization of the inner core, diamond exsolved and floated to the core-mantle boundary. Such a late diamond layer would have continued to grow throughout core crystallization.

lower mantle stopped at 3.7 Ga[44] and that inner crystallization only started at about 4 Ga[62]. Most of the diamond layer or its graphite precursor therefore formed after strong mantle convection stopped which limits the extent to which the diamond layer may have been disrupted and the extent to which graphite may have been redistributed by gravitational instability. Diamond is not expected to be affected by gravitational instability because it is denser than forsterite and enstatite, the main silicate minerals of the present-day mantle. Future work should investigate how the presence of a km-scale diamond layer at the CMB would affect the thermal evolution of the silicate fraction of Mercury[61]. We however believe that such a thin layer could not be unambiguously confirmed using interior models fitting Mercury's MOI and tidal Love numbers[30] given the current large uncertainties on the physical properties of Mercury's mantle and core[24,25,30].

Based on geodetic data, it has been suggested that a 100 km-thick FeS layer at Mercury's present day CMB may also exist[63]. The physical state (solid vs. liquid) of such a layer is unknown and experimental studies combined with geochemical data have shown that this layer, if present, should be much thinner than initially suggested from geophysics[10,64,65]. The occurrence of a diamond layer at the CMB is compatible with the presence of a FeS layer as both relate to the saturation of an element (C and S) during planetary differentiation. We propose that if the FeS layer is in a liquid state, the diamond layer would lie at the interface between the FeS layer and the silicate mantle. If the FeS layer is solid, the diamond layer would likely lie at the interface between the liquid outer metallic core and the solid FeS layer. Note however that given the strong density contrast between diamond (3500 kg/m³ and solid FeS 4840 kg/m³ in standard conditions) and the likely low viscosity of a hot, solid, FeS layer, gravitational instability may lead to overturn between the FeS layer and the diamond layer.

## Carbon cycle on a reduced terrestrial planet

Diamond is commonly reported in meteorites and is interpreted to have been transformed from graphite by shocks on their parent bodies[66]. Alternatively, diamond may grow up under high-pressure within parent bodies, as reported for ureilites[67], although this interpretation is debated[68]. It has been suggested that diamond formed during magma ocean crystallization on Earth[69,70], although diamonds found at the surface are metasomatic[71]. Exoplanets and their compositional diversities, including potential carbon-enrichment and high C/O ratios[72], may allow a variety of processes to produce diamond, from direct crystallization in the interior to impact processes at the surface[73], and hydrocarbons may even play a potential role[74,75]. On Mercury, diamond most likely formed during core crystallization and is now stored at the CMB, and it may have also formed in the deep magma ocean. Additionally, any graphite trapped in deep magma ocean cumulates may have transformed into diamond as the interior cooled. As discussed above, disruption of the diamond layer is only plausible during the early magmatic history of Mercury. As such we believe bringing those diamonds to the surface by remelting of the deep mantle could have happened during the production of high-Mg lavas that require deep melting sources[44]. This could potentially be investigated by detailed observation of the High-Mg province of Mercury by BepiColombo. At the surface, though, because of the high abundance of graphite from the primordial flotation crust, we consider impacts as the main process capable of forming any significant volume of diamond via the transformation of graphite[76].

These new views on the evolution of the carbon cycle on a terrestrial planet have implications for the deep storage of this life-forming volatile element[77,78]. Compared to other terrestrial planets, Mercury differentiated under unique reduced conditions that modify the behaviors of many chemical elements[79]. The partitioning of elements between reservoirs to form a Si-rich iron core and a sulfur-rich, iron-poor silicate portion makes Mercury unique. The peculiar conditions at the CMB, the formation of the solid inner core, and the saturation of the planet with carbon have likely resulted in the formation of a diamond layer at the CMB (Fig. 3). Whereas the previously proposed FeS layer at the CMB[29,63,80,81] would insulate the core from cooling[82], its unlikely occurrence[64,65] and the evidence provided here for the presence of a highly conductive diamond layer imply that heat transfer from the liquid outer core may favor thermal stratification at the top of Mercury's core, with potential implications for the generation of its magnetic field[83].

## Methods
### Mercury's core-mantle boundary
Mercury is in Cassini state 1[84,85], in which the obliquity of the body is directly related to its moments of inertia as:

$$K_1(\theta)\left(\frac{(C-A)}{C}\right) + K_2(\theta)\left(\frac{(B-A)}{C}\right) = K_2(\theta) \tag{1}$$

where $K_1$, $K_2$, and $K_3$ are constants involving the orbital eccentricity, inclination, mean motion, and planetary spin angular velocity, and explicitly the obliquity $\theta$. $A < B < C$ are the principal moments of inertia which are related to the amplitude of the 88-day librations ($\phi$) as:

$$\phi = \frac{3}{2}\frac{(B-A)}{C}\left(1 - 11e^2 + \frac{959}{48}e^4 + \ldots\right) \quad (2)$$

In Eq. 2, C should be replaced by $C_{m+cr}$, the value appropriate for the mantle (m) and crust (cr) if the mantle is decoupled from a molten core that is not involved in the 88-day librations.

The gravity field of a body of mass $M$ and radius $R$ can be described with spherical harmonics. Obliquity in combination with degree-2 gravity information thus provides a direct constraint on the polar moment of inertia $C$:

$$C_{20} = \frac{(C - (A+B)/2)}{MR^2} \quad (3)$$

$$C_{22} = \frac{(B-A)}{4MR^2} \quad (4)$$

Combining Eqs. 1, 3 and 4, we obtain:

$$\frac{C}{MR^2} = (-C_{20} + 2C_{22})\frac{K1(\theta)}{K3(\theta)} + 4C_{22}\frac{K2(\theta)}{K3(\theta)} \quad (5)$$

Based on the most recent estimates of MESSENGER orbital tracking data[24], Mercury's gravitational parameter (GM) is $(2.20318636 \pm 0.000000060) \times 10^{13}$ m$^3$ s$^{-2}$. With $G$, the gravitational constant, being $(6.67408 \pm 0.00031) \times 10^{-11}$ m$^3$ kg$^{-1}$s$^{-2}$, we obtain Mercury's mass to be:

$$(6)$$

The most recent gravitational field solution for Mercury (HgM008) places new constraints on $C_{20} = (0.80415 \pm 0.0003) \times 10^{-5}$ and $C_{22} = (0.80415 \pm 0.0002) \times 10^{-5}$. Different approaches have been followed by[24,26] to obtain the obliquity and libration amplitude of Mercury, providing different estimates of Mercury's normalized polar moment of inertia are $\frac{C}{MR^2} = 0.333 \pm 0.005$[24] and $0.343 \pm 0.006$[26].

We recalculated the pressure conditions at Mercury's core-mantle boundary (CMB) using the results of the Markov Chain Monte Carlo simulation of Mercury's internal layering[24].

The pressure at the CMB is calculated as:

$$P(r) = \int_0^r \rho(x)g(x)dx \quad (7)$$

with, $g$, the local gravity inside a sphere of radius r, defined as:

$$g(r) = \frac{G}{r^2}M(r) = \frac{G}{r^2}4\pi \int_0^r \rho(x)x^2dx \quad (8)$$

with $M(r)$ being the mass within the same sphere. For these calculations, we consider that the densities of the crust ($\rho_{cr}$) and the mantle ($\rho_m$) are constant throughout the thickness of these two layers. We calculated Mercury's CMB pressure conditions using the results of 439,000 simulations[25]. We obtained CMB pressures of $5.77 \pm 0.31$ GPa using the MOI of Genova et al. (2019) and $5.38 \pm 0.37$ GPa using Bertone et al.[26] (Supplementary Fig. S1).

## Experiments
**Starting compositions.** Synthetic silicate starting materials (Supplementary Table S1) were prepared from high-purity oxide powders:

$SiO_2$, $TiO_2$, $Al_2O_3$, $Cr_2O_3$, MnO, MgO, $CaSiO_3$, $Na_2SiO_3$, $K_2Si_4O_9$, and $AlPO_4$. These powders were mixed in appropriate proportions in an agate mortar. Silicon was added both as $SiO_2$ and metallic Si (Si/$SiO_2$ = 0.2) in order to reduce the starting material. Sulfur was added as FeS (20 wt%) to the starting silicate material (Table S1). A graphite capsule was used to ensure carbon saturation.

**Experimental methods.** Experiments were performed in a six-anvil cubic press with a maximum load of about 2700 tons on every WC anvil ($6 \times 27$ MN)[86]. The maximum oil pressure is about 110 MPa using the diameter of the individual ram of 0.56 m. To create high pressure, six WC anvils with a square-shaped truncation and driven from three perpendicular dimensions by a computer-controlled hydraulic system pressurize the central pyrophyllite cubic block ($38.5 \times 38.5 \times 38.5$ mm$^3$). The pyrophyllite cubic block serves as both the pressure medium and the gasket material (Supplementary Fig. S2). Cell pressures were determined using the phase transitions of Bi (I–II transition at 2.55 GPa and III–V at 7.7 GPa), Tl (II–III at 3.68 GPa), and Ba (I–II transition at 5.5 GPa[87,88]. The phase transitions were detected using the resistance measurement method[89]. For these experiments, a hand-machined graphite bucket with an I.D. of 0.7 mm, O.D. of ~1.7 mm, and a length of 3–4 mm, was filled with starting material and closed with a graphite lid. Temperature was monitored using a $W_{97}Re_3 - W_{75}Re_{25}$ (type-D) thermocouple and Eurotherm 2404 programmable controller. The capsule was placed in the hotspot of the assembly and adjacent to the thermocouple conjunction point (less than 1 mm) to minimize the thermal gradient and obtain accurate thermocouple readings; sample temperatures were within 5 K of the thermocouple reading[90]. Experiments were pressurized cold to the target pressure, then heated while maintaining pressure. Experimental pressures were $7 \pm 0.5$ GPa and temperatures ranged from 1973 to 2273 K. Experimental durations varied between 5 min and 6 h depending on the temperature and degree of melting. Upon completion of an experiment, runs were quenched by cutting power to the heater and the temperature typically dropped to below the glass transition within 5 s.

**Analytical methods.** Experimental run products were mounted in epoxy and dry polished with 1 μm diamond paste. The surfaces of polished specimens were carbon-coated for backscattered electron (BSE) imagery to assess the texture and mineralogy, and for electron microprobe analysis (EMPA) to obtain accurate chemical compositions for each phase. The chemical compositions of the run product phase (silicate minerals, silicate glasses, sulfides, and metallic alloys) were determined using a JEOL JXA-8230 Electron Microprobe at the Testing Center of Shandong Bureau of China Metallurgical Geology Bureau. Major and minor elements of all phases were deter- mined using a 15 kV accelerating voltage. A focused beam with a current of 20 nA was used to analyze silicate mineral phases. Large melt pools with quench textures were analyzed with a 20 nA beam defocused to a diameter of 20 μm. A defocused beam with a diameter size of 5–10 μm and 5–10 nA was used to analyze small melt pools in order to minimize beam damage and avoid alkaline migration. Sulfides and metallic alloys were analyzed with 15 kV and 20 nA focused beam. The standards were natural minerals and synthetic oxides for silicate phases (Si and Na, Jadeite; Ti, Rutile; Al, Garnet; Fe, Fayalite; Mg, Forsterite; Ca, Diopside; K, Sanidine; Cr, Chromite; Mn, Rhodonite; P, Apatite; S, pyrrhotite). Natural sulfide minerals and pure metals were used as standards for sulfide and metallic alloys to minimize the matrix effect. Peak and background counting times were between 10 and 20 s depending on the concentration of each element. Compositions reported here are based on 3–10 analyses per phase. The average compositions and standard deviations of analyzed phases of all experiments are shown in Supplementary Data 1. BSE images of representative run products are shown in Supplementary Fig. S3. Mineral and melt proportions were determined by mass balance using the EMPA data for run product

phases. Raman spectroscopic measurements were carried out using a Raman micro- scope equipped with a Renishaw inVia Raman spectrometer, a 532 nm laser, and a 20× objective. The instrument was routinely calibrated using a silicon standard. Analyses were performed using 2400 grooves mm$^{-1}$, an exposure time of 10 s, and 2 accumulations. The backscattered Raman radiation was collected on a polished sample surface over the range 100–2000 cm$^{-1}$ (Supplementary Fig. S4).

**Calculation of oxygen fugacity.** Oxygen fugacity ($fO_2$) in the experiments were calculated relative to the iron-wüstite (Fe-FeO; see below) using the following expression:

$$\Delta IW = 2\log(a_{FeO}^{silicate}/a_{Fe}^{metal}) \tag{9}$$

Where $a_{FeO}^{silicate}$ and $a_{Fe}^{metal}$ are the activities of FeO in the silicate melt and Fe in the molten metal, respectively. Activities should be calculated using the activity coefficient ($\gamma$) and the mole fraction ($\chi$). In this study, we however assumed ideal solutions so that $\gamma_{FeO} = 1.0$ and $\gamma_{Fe} = 1.0$.

## Mercury's magma ocean liquidus

We calculated the most likely distribution of Si in Mercury's core using the results of Goossens et al.[25] to be 5.61 ± 3.11 wt% ($x \pm 1\sigma$) using MOI = 0.333 ± 0.005[24] and 12.15 ± 4.31 wt% ($x \pm 1\sigma$) using MOI = 0.343 ± 0.006[26]. We thus calculated the liquidus curve of three S-free bulk silicate compositions (EH-EL (BSM), with no Si in the core; SiCore$_4$, with 4 mol% Si in the core; and SiCore$_8$, with 8 mol% Si in the core; Supplementary Table S1). We tracked the liquidus curve using the thermodynamic model of Holland et al.[43] as implemented in the MAGEMin Gibbs free energy minimization software[42].

Calculations were performed over six successive iterations to track the boundaries between melt and forsterite and/or enstatite. Following the approach of[91], we parameterized the liquidus equations as:

$$T_{liq}, BSM = 2092 + 3.40P + 16.88P^2 - 2.43P^3 + 0.14P^4 \tag{10a}$$

$$T_{liq}, BSM - SiCore_4 = 2106 - 7.46P + 25.93P^2 - 4.47P^3 + 0.27P^4 \tag{10b}$$

$$T_{liq}, BSM - SiCore_8 = 2124 + 5.58P + 14.28P^2 - 2.23P^3 + 0.14P^4 \tag{10c}$$

The calculated liquidus curves from 0 to 8 GPa are shown in Supplementary Fig. S5. Comparison of these calculated liquidus curves for S-free compositions allows us to estimate the impact of S on depressing the liquidus. At 7 GPa, the pressure of the experiments performed in this study, the three S-free bulk compositions had very similar liquidus temperatures of ∼2450 K. At the same pressure, our experiments with 6 wt% S had a liquidus temperature of ∼2215 K. Because pressure has no significant effect on sulfur solubility in reduced magmas[10], a single non-linear expression is used to define the liquidus depression per wt% S as:

$$\left[\frac{dT}{dS}\right]_P = 59.2X_S^{0.75} \tag{11}$$

In Eq. 11, we used a power factor identical to that determined for H$_2$O[92,93]. The liquidus depression trend that we calculate is of the same order of magnitude as that determined for the effect of H$_2$O on the olivine or plagioclase liquidi in basalt[92,94] (Supplementary Fig. S6).

## Graphite–diamond transition

Graphite and diamonds are two polymorphs of elemental C; diamond is the high-pressure polymorph whereas graphite is the low-pressure polymorph. The transition between the two phases was calculated using the model of Day[45] for the reaction:

$$C^{Graphite} = C^{Diamond} \tag{12}$$

The Gibbs free energy of reaction (Eq. 12) in standard conditions ($\Delta G_r^o$) is the enthalpy of formation ($\Delta H_f^o$) and entropy ($S^o$):

$$\Delta G_r^o = \Delta H_f^o - T\Delta S^o \tag{13}$$

Enthalpy and entropy terms were calculated at the temperature of the reaction using a Maier-Kelly polynomial expression of the heat capacity ($C_P$):

$$CP = a + bT + cT^{-2} + dT^{-0.5} + eT^2 + fT^{-3} + gT^{-4} + hT^3 \tag{14}$$

The contribution to the Gibbs free energy due to variations in pressure ($V\delta P$) is calculated using an equation of state following the formalism of Murnaghan[95]:

$$\frac{V}{V_0} = \left(1 + \frac{n}{B_0}P\right)^{-\frac{1}{n}} \tag{15}$$

where $V$ is volume, $V_0$ is the volume at zero pressure, $P$ is pressure, $B_0$ is the initial bulk modulus and $n$ is the pressure derivative of the bulk modulus.

For both phases, $V$ is calculated at a given temperature using expressions of thermal expansion:

$$(V/V_0)_{Gr} = 1 + \alpha_o(T - 298) - 20\alpha_o(T^{0.5} - 298^{0.5}) \tag{16a}$$

$$(V/V_0)_{Dia} = \exp(3\sum[X_i\theta_i/\exp(\theta_i/T) - 1]) \tag{16b}$$

All thermodynamic parameters used in Eq. 13, 14, 15, 16a and 16b are given in Day[45].

## Carbon in Mercury's mantle

**Carbon solubility.** We calculated the solubility of C in Mercury's mantle, i.e., the maximum C content at graphite or diamond saturation, using two models. In the first, we used the experimental regression of Li et al.[19]:

$$\log C[ppm] = 0.96\log X_{H_2O} - 0.25\Delta IW + 2.83 \tag{17}$$

with $\Delta IW$ being the oxygen fugacity ($fO_2$) relative to the Fe–FeO buffer. We calculated the $fO_2$ of IW using the Gibbs free energy of formation ($\Delta G_f^o$) of FeO listed in the NIST-JANAF Thermochemical Tables. We considered $XH_2O = 0.01$.

In the second model, we calculated the solubility of C based on the following two thermodynamic equilibria:

$$2C + O_2 = 2CO \tag{18a}$$

$$C + O_2 = CO_2 \tag{18b}$$

We calculated $fCO$ and $fCO_2$ as a function of $fO_2$ using the $\Delta G_f^o$ values for CO and CO$_2$ listed in the NIST-JANAF Thermochemical Tables. Fugacities were then linked to the total C content of Mercury's magma ocean using the expression:

$$C[ppm] = K_{CO}fCO + K_{CO_2}fCO_2 \tag{19}$$

where $K_{CO}$ and $K_{CO_2}$ are Henry's constants of CO and $CO_2$ provided by Keppler and Golabek[20]. Results of the two calculation methods are provided in Supplementary Fig. S10.

Using the model of Li et al.[19] and considering that Mercury differentiated at a $fO_2$ between IW-5 and IW-6[10], we calculate a total carbon content of 7-15 ppm C in the magma ocean. Thermodynamic calculations of CO and $CO_2$ solubility[20] are 1 or 2 order of magnitudes lower (0.06 to 0.2 ppm). We are not aware of any study evaluating the effect of C content on the liquidus of mafic magmas. However, by analogy with $H_2O$[93], we calculate that C depresses the liquidus by less than 1 K. This is obviously insignificant on comparison to the large effect of S on the liquidus depression.

**Thickness of graphite or diamond layers from the magma ocean**

We calculated the thickness that a floated graphite layer would have if all carbon dissolved in the magma ocean formed graphite during cooling and crystallization. For these calculations, we considered a core radius of 1950 km[25], a magma ocean density ($\rho$) of 3000 kg m$^{-3}$, and a graphite density of 2100 kg m$^{-3}$ [21]. Results are shown in the right panel of Supplementary Fig. S7.

We also hypothesized that the deepest part of Mercury's magma ocean might crystallize diamond instead of graphite. We considered that only the lowest ~79 km of the magma ocean, corresponding to 4.77–5.77 GPa, could be in the stability field of diamond. Using a diamond density of 3500 kg m$^{-3}$ and considering that diamond would accumulate at the interface between the liquid core and the magma ocean, we calculated the thickness of the diamond layer as shown in Supplementary Fig. S8.

**Carbon in FeSi metal**

The carbon content in carbon-saturated FeSi metal was recently investigated[13] based on new high-temperature, low-pressure experiments and a review of the metallurgical literature. We extended this review by compiling 598 experiments containing metal in C-saturated systems. We ultimately used only 244 experiments with metal containing 0.25–25 wt% Si to ensure that all metals were well within the Fe-Si-C system and had C and Si concentrations above the detection limits for bulk and in situ analytical methods. Using these experiments, we observed a coherent trend of decreasing bulk C content with increasing Si content in the metal (Supplementary Fig. S9). We do not observe any temperature-dependency on C solubility in FeSi metal as already suggested[13]. We therefore modeled C solubility [wt%] as a function of the metal Si content [wt%] as:

$$C_{Fe-Si} = 5.48 - 0.45Si + 0.009Si^2 \qquad (20)$$

Using Eq. 20 and the geophysical results[25], we defined the distribution of plausible C concentrations in the metallic alloy if C-saturated (Supplementary Fig. S10). We also consider the incorporation of carbon in the FeSi solid inner core by using a liquid/solid partition coefficient of 0.3[52].

## Data availability

The experimental data are provided in the Supplementary Information Data 1 and deposited in a publicly available Zenodo repository (https://zenodo.org/records/11107570). The data for interior structure model[25] used in this study are available in the NASA archive (https://pgda.gsfc.nasa.gov/products/83).

## Code availability

The MAGEMin thermodynamic calculator[45] used to perform all simulations of liquidus temperature, MAGEmin is publicly available at Github: https://github.com/ComputationalThermodynamics/MAGEMin. The Python script used for the thermodynamic calculations is publicly available and has been deposited in Zenodo repository (https://zenodo.org/records/10839707).

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

## Acknowledgements

This research was supported by a grant from the National Natural Science Foundation of China awarded to Y.L. (42250105). The Center for High Pressure Science and Technology Advanced Research is supported by the National Science Foundation of China (Grants U1530402 and U1930401). BC is a Research Associate of the Belgian Fund for Scientific Research-FNRS and acknowledges funding from the ESA PRODEX Program (Grant 4000142722).

## Author contributions

Y.L. and B.C. conceptualized and designed the project. Y.X., P.W., and Y.L. performed the experiments and sample analyses. O.N., Y.Z., and B.C. performed the interior model and thermodynamic calculations. All authors discussed the results and wrote the paper.

## Competing interests

The authors declare no competing interests.
