## [Peer Review File · Nature Communications]

Editorial Note: Figure on page 10 of this Peer Review File have been redacted as indicated to remove third-party material where no permission to publish could be obtained.

REVIEWER COMMENTS

Reviewer #2 (Remarks to the Author):

Review of "A diamond-bearing core-mantle boundary on Mercury" by Xu et al. submitted to Nature Communications

This manuscript is well written and organized. The authors present new high P-T experimental data on the liquidus of Mercury's magma ocean analog combined with thermodynamic modeling and revised geodetic calculations to evaluate the stability of graphite/diamond at Mercury's core-mantle boundary (CMB) conditions. The authors claim that with sulfur as a major element in the magma ocean, it is likely although not statically favorable that diamond could have crystallized at the bottom of the magma ocean. In addition, the authors also suggest that the crystallization of a FeSi solid inner core would exsolve C as diamond, which would float to CMB and become stable with secular cooling. This study definitely presents a very interesting story that may significantly improve our understanding of the internal structure and evolution of Mercury, particularly for diamond formation processes on planetary bodies, and is expected to interest a broad range of readers in Earth and planetary science. While I recognize the importance and novelty of this study, I do have some concerns about the experimental details, as well as some interpretations of the data. As such, I think this paper would be suitable for publication in Nature Communications, given that the authors could address the comments raised below

Detailed comments:

(1) As stated throughout the whole manuscript, the authors assumed that the early magma ocean and the core of Mercury were saturated with carbon. However, the starting materials for the experiments simulating Mercury's magma ocean do not have any carbon (Table S1). So I am a bit confused, how the liquidus T of a magma ocean with compositions that could not crystallize any carbon-bearing phases can be used to place constraints on the occurrence and thermodynamic stability of diamond during the magma ocean crystallization? I might have missed something in the manuscript, but the authors have not clearly discussed this in the experimental section. The authors added FeS to the starting materials to account for the potential of sulfur as a major element in the magma ocean due to its abundance on Mercury's surface, although the exact amount of added FeS is not given in the manuscript. But for carbon, it was not clearly discussed. Although carbon solubility in silicate melts may be low, the magma ocean should have some in order to generate graphite/diamond. And also, whether the presence of trace amount of carbon can affect the liquidus of the magma ocean also needs to be discussed. Overall, I think the choice of starting materials needs more justification if a revision is extended by the editor.

(2) The experiments were performed only at 7 GPa, which is higher than the estimated pressure range at CMB, I am curious why the authors chose a higher pressure. Higher pressure tends to favor the stabilization of diamond, and may also affect the solubility of sulfur as well, would this introduce any bias in the model without any relatively lower pressure experimental data point? Speaking of sulfur content, I couldn't find in Figure 1 or Table S1 the sulfur content in the experiments, neither the amount of S added to the starting materials nor the measured S content in the quenched melt. As these are critical parameters for evaluating the effect of sulfur on the liquidus depression, they should be reported either

in the table or labelled in the figure for each experiment.

(3) Related to the above comment, it seems like the experimental results were mainly used to estimate the effect of sulfur on the liquidus depression at 7 GPa, compared with sulfur-free liquidus obtained from thermodynamic models. I am wondering why the authors did not perform an experiment on sulfur-free composition so that the effect of sulfur can be evaluated in a more consistent way, and then the thermodynamic model can also be compared with and calibrated by the sulfur-free experiments. To me, thermodynamic models always have inherent uncertainty that might be different than those from the experiments, particularly if the model has not been calibrated by compositions and conditions relevant to Mercury's interior. So using the modeled liquidus as a basis for evaluating experimental results, although still useful, is probably less convincing if the calculated liquidus has not been compared with experiment.

(4) As I am an experimentalist, I might be a bit picky about the experimental details presented in this paper. I am a bit surprised that the authors could achieve ~ 7 GPa using such a large cell assembly (~ 52.5 mm). A large cell assembly will inevitably have large thermal gradient across the sample. I wonder if the authors have calibrated the thermal gradient in the cell, either by experiments or thermal modeling? Since the reported T is close to the boundary of graphite/diamond transition and all the rest discussions rely on the degree of liquidus depression, accurate determination of the T within the sample is crucial. And also, please provide a schematic drawing of the cell assembly used for the experiments, which would help the readers easily get the information on the experiments. In addition, the authors mentioned the pressures were calibrated by several phase transitions, but those phase transitions listed in the paper are all for room-T calibration. Has any high-T phase transition been used in the calibration? As the current experimental T is high (>2000 K), the pressure change due to thermal pressure and thermal relaxation of the cell at high T would be significant. If the pressure was only estimated from room-T calibration, it might significantly deviate from the estimated room-T value at high T. Such uncertainties also need to be considered when evaluating the P-T conditions of the magma ocean.

(5) In terms of the carbon content in Mercury's mantle, it looks like the two approaches the authors employed predict totally different f_{O_2} dependence of the carbon content, as well as the resulting thickness of the graphite/diamond layer (e.g., Fig. S8). I wonder if the authors could compare the estimated carbon content in the magma ocean with the measured carbon content in the crust, not just the thickness, assuming all carbon in the magma ocean turned into graphite in the crust. Since we have a relatively good knowledge of the abundance of carbon in the crust, such comparison may help evaluate which model predicts the carbon content more precisely. And, we can also evaluate from this comparison whether the carbon exsolution from the core, according to the authors' hypothesis, might have also contributed to the graphite crust formation in the early stage when the T at CMB was high.

Some minor issues:

(1) Quite a few figures and tables are cited incorrectly in the main text. The authors should check the order of their figures/tables, and make sure they are mentioned correctly in the appropriate position in the text.

For example,

Line 93: Table S1 does not provide any info on the oxygen fugacity
Line 136: I think Fig. S8 here should be Fig. S9
Line 139: Fig. S7 should be Fig. S8
Line 147: Fig. S5 should be replaced by Fig. S6
Line 155: I don't know which figure the authors refer to (Fig. S7?), but there's no Fig. 2a in the manuscript.
Line 305: No Table S2 (Appendix) provided
Line 321 and 330: Table 2 should be Table S2 in the Supplementary Materials
Line 452: Is the Supplementary Table S1 refer to the attached excel file which the authors sometimes refer as Appendix? Please unify the naming of all supporting figures and tables in a consistent way to avoid confusion.

(2) Figure S10, the legend in the figure and the description in the caption do not match

(3) Lines 127-130: I think a figure is needed for this description

(4) Line 196: I think a word is missing here. It should be "reduced conditions"

(5) Table S2. This table reports sulfur-free magma ocean compositions used in the thermodynamic modeling, and Table S1 also reports sulfur-free compositions used in the experiments (the added amount of FeS is not given!). So I suppose the composition for Mer8 in Table S1 should be similar to that of BSM-SiCore8 in Table S2. I think it might be better to merge the two tables into one and use the same unit for the composition or report both mol% and wt% to facilitate comparison.

(6) Line 380: Please check the equation to make sure it's correct

(7) The excel file Table S1 Summary of experimental results. I think the added FeS amount for each experiment should also be listed. A quick look at the table shows that the glass composition with the highest sulfur content (~16.23%, run Mer8-2) is significantly different than the glass composition in other experiments (much lower MgO content, significantly higher CaO, Na₂O and K₂O content), probably due to its small fraction causing difficulty in analyzing? Would this affect the interpretation of the sulfur effect on melt liquidus?

Reviewer #3 (Remarks to the Author):

Summary:

The manuscript Xu et al. presents novel high-temperature and high-pressure experiments and liquidus thermodynamic modelling to understand the stability of graphite vs. diamond, finding diamond was possibly, though statistically unlikely, to be stable in Mercury's silicate magma ocean, but would be predicted to form a thick layer at the current core-mantle-boundary (CMB). A conductive diamond layer on top of the CMB would have significant implications for Mercury's thermochemical evolution given its opposite properties to an unlikely insulating FeS layer. The manuscript is extremely well written and thorough in its incorporation of literature data and plentiful supplementary figures. I recommend

acceptance.

Quantifying the effect of S on melting T in itself is a super interesting result worth highlighting and further study (either adding to the main text or in another manuscript rather than hidden in the supplementary). Please expand on the form of equation 12 to define the liquidus depression per wt% S. It is interesting that it is on the same order of magnitude of the effect of H₂O on olivine or plagioclase liquid in basalt. Would be interesting to comment on whether the S effect changes with pressure too.

Reviewer #4 (Remarks to the Author):

In this manuscript, Xu et al. explore the possibility of a diamond-bearing layer at the core-mantle boundary in Mercury. They utilized experimental techniques to understand the geochemistry and phase stability of Mercury relevant compositions, paring the results with geophysical models to estimate the dynamics of diamond growth and mobility in the mantle and core. This is a novel idea and has the potential to have an impact on Mercury science. However, there are several major points that I believe need to be addressed before this can be suitable for publication.

-Other work has suggested that there would be an FeS layer at the core-mantle boundary (CMB) (e.g., Harder & Schubert, 2001; Smith et al., 2012; Hauck et al., 2013) as a way to explain the moment of inertia (MOI) and the abundance of S on the surface. This seems to be taking a similar approach, just from the perspective of C. Is this an alternative to the FeS hypothesis? Do these analyses fit with the MOI measurements better than a possible FeS layer? I think evidence as to how this is a more plausible hypothesis (if this is indeed the case) will strengthen this work.

[<https://doi.org/10.1006/icar.2001.6586> <https://doi.org/10.1126/science.1218809>
<https://doi.org/10.1002/jgre.20091>]

-Please provide a pressure scale on Figure 3.

-Regarding lines 152-154, are the diamonds growing and mobilizing from what is interpreted to be the liquid outer core or are diamonds co-crystallizing as the core solidifies? Clarification would be helpful. Related to this point, at what timescale would this occur at and when would a diamond layer be expected to be present at the CMB?

-In the section starting at Line 70, the CMB pressure is calculated to be 5.77 GPa and in lines 108-109 it is stated that the pressure at the bottom of Mercury's magma ocean was similar to present CMB, but the experiments are run at 7 GPa. What is the rationale for this- why weren't experiments conducted at a pressure that is more similar to the predicted CMB pressure calculated in this work?

-At Mercurian CMB pressure conditions and a range of relevant temperature (early and present) this is sitting at metastability for diamond and graphite which is mentioned in line 176. I am not convinced at the possible longevity of this layer, lasting long enough to be measurable via MOI today. If some of the layer were to transform to graphite, it would be gravitationally unstable and reduce the thickness of the layer, thus dampening the effect of this diamond bearing layer. Additionally, convection in the first Ga of

Mercury's history could disrupt a diamond layer and redistribute in the mantle.

-The density of the solidified magma ocean and diamonds needs to be scaled with pressure. With the minerals being considered (e.g., opx, cpx, garnet, olivine) the density is likely greater than the 3000 kg/m³ being considered when solidified. This could be an issue thinking about gravitational stability of diamond below a potentially denser silicate at the CMB after the magma ocean stage.

-Figure S2a- it looks like there is a mid-gray phase surrounding the large phase that is labeled "Sf" (and some of the smaller sulfides). In experiments that are rich in Fe and S that are at lower temperatures (which S2a is the lowest T) there is phase separation of a sulfide-rich rim with a metal (Fe)-rich core, evolving to a single phase with increasing temperature. Is this the case for this experiment that there are indeed two phases? If so, it would be beneficial to label the two phases appropriately.

-Lines 275-275- what assembly was used (e.g., 14/8)? Were WC cubes used for these experiments?

Response to reviewer comments

[Referee comments in *italics*, responses in normal font and blue]

Reviewer #2 (Comments for the Author):

Review of “A diamond-bearing core-mantle boundary on Mercury” by Xu et al. submitted to Nature Communications

This manuscript is well written and organized. The authors present new high P-T experimental data on the liquidus of Mercury’s magma ocean analog combined with thermodynamic modeling and revised geodetic calculations to evaluate the stability of graphite/diamond at Mercury’s core-mantle boundary (CMB) conditions. The authors claim that with sulfur as a major element in the magma ocean, it is likely although not statically favorable that diamond could have crystallized at the bottom of the magma ocean. In addition, the authors also suggest that the crystallization of a FeSi solid inner core would exsolve C as diamond, which would float to CMB and become stable with secular cooling. This study definitely presents a very interesting story that may significantly improve our understanding of the internal structure and evolution of Mercury, particularly for diamond formation processes on planetary bodies, and is expected to interest a broad range of readers in Earth and planetary science. While I recognize the importance and novelty of this study, I do have some concerns about the experimental details, as well as some interpretations of the data. As such, I think this paper would be suitable for publication in Nature Communications, given that the authors could address the comments raised below.

Reply: We appreciate that reviewer #2 recognizes the importance and novelty of our study and for recommending publication of our paper after revision. Below we provide detailed replies to all points the reviewer raises. Comments made by reviewer #2 enable us to explain more carefully the experimental approach and the relevance of P-T conditions we investigated. However, the main conclusions of our study remain unchanged.

Detailed comments:

(1) As stated throughout the whole manuscript, the authors assumed that the early magma ocean and the core of Mercury were saturated with carbon. However, the starting materials for the experiments simulating Mercury’s magma ocean do not have any carbon (Table S1). So I am a bit confused, how the liquidus T of a magma ocean with compositions that could not crystallize any carbon-bearing phases can be used to place constraints on the occurrence and thermodynamic stability of diamond during the magma ocean crystallization? I might have missed something in the manuscript, but the authors have not clearly discussed this in the experimental section. The authors added FeS to the starting materials to account for the potential of sulfur as a major element in the magma ocean due to its abundance on Mercury’s surface, although the exact amount of added FeS is not given in the manuscript. But for carbon, it was not clearly discussed. Although carbon solubility in silicate melts may be low, the magma ocean should have some in order to generate graphite/diamond. And also, whether the presence of trace amount of carbon can affect the liquidus of the magma ocean also needs to be discussed. Overall, I think the choice of starting materials needs more justification if a revision is extended by the editor.

Reply: We thank the reviewer for pointing this out. A graphite capsule was used in our experiments to ensure carbon saturation in the starting material. The addition of carbon to the starting powder is thus not necessary. The use of a graphite capsule was already mentioned in the caption of Figure 1 and reported also in the Methods section (in Experimental methods). Nevertheless, we added the sentence “A graphite capsule was used to ensure carbon saturation.” to the section on “Methods – Starting composition”.

The exact amount of FeS added to the starting materials has also been reported in “Methods – Starting composition”. We also added the relative fraction of Si and SiO₂ used in the starting silicate material. Regarding the effect of carbon on the liquidus of the magma ocean, we provided a detailed model for carbon solubility in the silicate melt in the section on “Carbon in Mercury’s mantle – Carbon solubility”. In the revised version, we now provide additional information on the concentration of carbon to be expected in the magma ocean and show that those very low values cannot have any significant effect on the liquidus of the magma ocean as compared to the effect of sulfur. The following paragraph has been added at the end of the section “Carbon in Mercury’s mantle – Carbon solubility”.

Using the model of ¹⁹ and considering that Mercury differentiated at a fO_2 between IW-5 and IW-6¹⁰, we calculate a total carbon content of 7-15 ppm C in the magma ocean. Thermodynamic calculations of CO and CO₂ solubility²⁰ are 1 or 2 order of magnitudes lower (0.06 to 0.2 ppm). We are not aware of any study evaluating the effect of C content on the liquidus of mafic magmas. However, by analogy with H₂O⁹⁴, we calculate that C depresses the liquidus by less than 1K. This is obviously insignificant on comparison to the large effect of S on the liquidus depression.

In the main text we added the following sentence: ‘Carbon solubility in the reduced magma ocean is orders of magnitude lower than that of sulfur^{10,14} which implies that carbon has no effect on the magma ocean liquidus temperature.’

¹⁰ Namur, O., Charlier, B., Holtz, F., Cartier, C. & McCammon, C. Sulfur solubility in reduced mafic silicate melts: Implications for the speciation and distribution of sulfur on Mercury. *Earth and Planetary Science Letters* **448**, 102-114, doi:<http://dx.doi.org/10.1016/j.epsl.2016.05.024> (2016).

¹⁹ Li, Y., Dasgupta, R. & Tsuno, K. Carbon contents in reduced basalts at graphite saturation: Implications for the degassing of Mars, Mercury, and the Moon. *Journal of Geophysical Research: Planets* **122**, 1300-1320, doi:<https://doi.org/10.1002/2017JE005289> (2017).

²⁰ Keppler, H. & Golabek, G. Graphite floatation on a magma ocean and the fate of carbon during core formation. *Geochemical Perspectives Letters* **11**, 12-17, doi:<https://doi.org/10.7185/geochemlet.1918> (2019).

⁹⁴ Katz, R. F., Spiegelman, M. & Langmuir, C. H. A new parameterization of hydrous mantle melting. *Geochem. Geophys. Geosyst.* **4**, 1073, doi:10.1029/2002gc000433 (2003).

(2) The experiments were performed only at 7 GPa, which is higher than the estimated pressure range at CMB, I am curious why the authors chose a higher pressure. Higher pressure tends to favor the stabilization of diamond, and may also affect the solubility of sulfur as well, would this introduce any bias in the model without any relatively lower pressure experimental data point? Speaking of sulfur content, I couldn't find in Figure 1 or Table S1 the sulfur content in the experiments, neither the amount of S added to the starting materials nor the measured S content in the quenched melt. As these are critical parameters for evaluating the effect of sulfur on the liquidus depression, they should be reported either in the table or labelled in the figure for each experiment.

Reply: 7 GPa is actually not higher than the estimated pressure range at CMB, but corresponds to the maximum value of potential pressures predicted by models of Goossens et al. (2022) using the normalized polar moment of inertia of Genova et al. (2019) (the model actually predicts values up to 7.19 GPa). It is true that pressure tends to favor the stabilization of diamond (at constant temperature). But a higher pressure also implies a higher liquidus temperature of the silicate melt which does not favor diamond stability (but favors graphite stability).

Running experiments at 7 GPa does not introduce any bias to the main conclusion of the paper because our results are based not only on experiments but on the combination between experimental results and careful modelling to cover the full P-T range of CMB conditions. It is

true that running experiments at 7 GPa enabled us to test the most favorable experimental conditions to produce diamond. This does not imply that our conclusions are forced to support diamond stability because our experimental results are implemented into models to interpolate the diamond-graphite stability for the whole range of potential CMB pressures predicted by models of Goossens et al. (2022).

As shown in Figure 1a of the paper, the temperature of the diamond-graphite transition is also close to the liquidus of the two starting compositions (Mer8 and Mer15). Additionally, a suite of experiments on sulfur solubility under reduced conditions were already available at pressure <4GPa, enabling us to interpolate any results between our experimental pressure (7 GPa) and those of Namur et al. (2016).

We added this sentence to the main text to make this clear: “This pressure corresponds to the maximum value of potential CMB pressures predicted by interior models²⁵.”

We changed the first part of the result section by adding the following sentence: “We used planetary interior structure models that satisfy the various measurements of MOI25 to calculate that the pressure at Mercury’s CMB is 5.77 ± 0.31 GPa with a low MOI24 and 5.38 ± 0.37 GPa with a high MOI26 (Methods and Supplementary Fig. S1). According to our calculations, the highest possible pressure of Mercury’s CMB is 7 GPa. We here provide experimental melting relations at 7 GPa to determine experimentally the carbon speciation at the most extreme pressure conditions of Mercury’s magma ocean. Experiments are then combined with thermodynamic modelling to investigate how phase relations change for lower pressure CMB conditions which would still match Mercury’s MOI.”

The role of pressure on sulfur solubility was discussed in Namur et al. (2016) who did not identify any significant effect as compared to oxygen fugacity, temperature, and melt composition. There is thus no reason to suspect that a slightly lower pressure (e.g. 5.5 GPa vs. 7 GPa) would have any effect on the conclusions of this study. Anyway, the effect of pressure on the liquidus temperature is considered in this study, as well as the effect of sulfur content. Any effect of pressure on sulfur solubility would not affect the model.

We have slightly modified the manuscript to make this clear by adding “Because pressure has no significant effect on sulfur solubility in reduced magmas¹⁰, a single non-linear expression is used to define the liquidus depression per wt% S”.

In the main text we added the following sentences: “It has been shown previously that sulfur solubility in reduced silicate melts is not pressure sensitive¹⁰ so that the liquidus depression we calculate as a function of sulfur content is independent on pressure conditions.”

Finally, the S content for each experiment was reported in the Table S1 of the submitted manuscript. We do not know if the format was changed from an excel table to a pdf version. We apologize for this potential confusion.

(3) Related to the above comment, it seems like the experimental results were mainly used to estimate the effect of sulfur on the liquidus depression at 7 GPa, compared with sulfur-free liquidus obtained from thermodynamic models. I am wondering why the authors did not perform an experiment on sulfur-free composition so that the effect of sulfur can be evaluated in a more consistent way, and then the thermodynamic model can also be compared with and calibrated by the sulfur-free experiments. To me, thermodynamic models always have inherent uncertainty that might be different than those from the experiments, particularly if the model has not been calibrated by compositions and conditions relevant to Mercury’s interior. So using the modeled liquidus as a basis for evaluating experimental results, although still useful, is probably less convincing if the calculated liquidus has not been compared with experiment.

Reply: This comment by reviewer #2 seems fully relevant at first sight but is confronted with the reality of the experimental approach that would not enable us to test experimentally the difference in temperature of the liquidus for a sulfur-free vs. a sulfur-bearing experiments.

In our study, we did not produce experiments just at the liquidus of the starting compositions. Instead, we have produced a suite of experiments for two starting compositions over a range of temperature. For each of those experiments, we have obtained a silicate glass with a composition varying with temperature (and the degree of crystallinity). In order to obtain the liquidus of each specific sulfur-free compositions, it would have been necessary to synthesize a new starting composition for each experiment (18 in total). For each of those starting compositions, it would have been necessary to run at least 4 experiments at different temperatures to identify precisely the liquidus, meaning that 72 experiments would have been necessary to follow the approach recommended by reviewer #2. This is unrealistic.

We decided to use the liquidus temperature of sulfur-free compositions predicted by the MAgEMin thermodynamic calculator because, in a previous study by Namur et al. (2016), it was clearly proved that liquidus temperature predicted by available Gibbs free minimization models give accurate results (within $\pm 21^\circ\text{C}$ (see figure below from the supplementary materials of Namur O, Collinet M, Charlier B, Grove TL, Holtz F, McCammon C (2016) Melting processes and mantle sources of lavas on Mercury. *Earth and Planetary Science Letters* 439:117-128 doi:<http://dx.doi.org/10.1016/j.epsl.2016.01.030>). As compared to a liquidus depression of ca. $250\text{-}350^\circ\text{C}$ for sulfur-bearing Mercury-like compositions, the error resulting from thermodynamic models is less than 10% of the absolute temperature depression. This minor error would not justify the need for running sulfur-free experiments.

[Redacted]

We added the following sentence in the manuscript: ‘Thermodynamic models have previously been shown to be very accurate to reproduce phase equilibria of sulfur-free Mercury-like magmas⁴⁵.’

(4) As I am an experimentalist, I might be a bit picky about the experimental details presented in this paper. I am a bit surprised that the authors could achieve ~7 GPa using such a large cell assembly (~52.5 mm). A large cell assembly will inevitably have large thermal gradient across the sample. I wonder if the authors have calibrated the thermal gradient in the cell, either by experiments or thermal modeling? Since the reported T is close to the boundary of graphite/diamond transition and all the rest discussions rely on the degree of liquidus depression, accurate determination of the T within the sample is crucial. And also, please provide a schematic drawing of the cell assembly used for the experiments, which would help the readers easily get the information on the experiments. In addition, the authors mentioned the pressures were calibrated by several phase transitions, but those phase transitions listed in the paper are all for room-T calibration. Has any high-T phase transition been used in the calibration? As the current experimental T is high (>2000 K), the pressure change due to thermal pressure and thermal relaxation of the cell at high T would be significant. If the pressure was only estimated from room-T calibration, it might significantly deviate from the estimated room-T value at high T. Such uncertainties also need to be considered when evaluating the P-T conditions of the magma ocean.

Reply: We thank the reviewer for pointing these out. In general, a higher temperature is observed at the center of the experimental assemblage and gradually decreases towards the edge in a commonly used 14/8 assembly in a multi-anvil press. The larger assembly we used in a Cubic press (a bit different than a common multi-anvil press) does not necessarily result in a thermal gradient through the sample. Actually, a larger assembly has thicker thermal insulation material than the assembly used in a common multi-anvil press, which can efficiently prevent the heat loss from the center and create a smaller thermal gradient at the hot-spot region. In our assembly, the graphite capsule itself serves as a heater, and the thermocouple junction penetrates the heater and is adjacent to the sample capsule. The thermal gradient is generally thought to be insignificant. We provide a schematic drawing of the cell assembly used for our experiments as a new Fig. S2. A detailed description of this assembly is shown in Fig. 3 in Wu et al. (2024) [this paper has just been published in Wu P, Xu Y, Lin Y (2024) A novel rapid cooling assembly design in a high-pressure cubic press apparatus. *Matter and Radiation at Extremes* 9, 027402. doi:10.1063/5.0176025

Fig. S2: Cubic press experimental assembly shown in vertical section, cross section, and 3D model. A slim graphite rod (2.5 mm in diameter) is placed vertically in the center of the assembly, and sample capsules are inserted into this rod. In addition, two sample capsules can be done at the same time, and the thermocouple junction is placed between the capsules in the center of the assembly. The length of cube edge is 38.5 mm.

We also did the pressure calibration at high-T with the phase transition diamond-graphite at 7 GPa (Day, 2012; A revised diamond-graphite transition curve, American Mineralogist). In our manuscript., the dashed line in Fig. 1a shows that the diamond-graphite transition is constrained at 2348 ± 25 K and 7 GPa by our experiments, which is consistent with the temperature of diamond-graphite transition at 7 GPa in the Fig. 7 shown in Day (2012). This demonstrates that our experimental pressure calibration is reliable. We clarify this in lines 108-111.

The vertical dashed line at 2356 K is the diamond-graphite transition at 7 GPa⁴², consistent with Raman spectra of ‘graphite’ capsules in our experiments at 2323 and 2373 K (Supplementary Fig. S4), demonstrating that the experimental pressure is reliable.

(5) In terms of the carbon content in Mercury’s mantle, it looks like the two approaches the authors employed predict totally different fO_2 dependence of the carbon content, as well as the resulting thickness of the graphite/diamond layer (e.g., Fig. S8). I wonder if the authors could compare the estimated carbon content in the magma ocean with the measured carbon content in the crust, not just the thickness, assuming all carbon in the magma ocean turned into graphite in the crust. Since we have a relatively good knowledge of the abundance of carbon in the crust, such comparison may help evaluate which model predicts the carbon content more precisely. And, we can also evaluate from this comparison whether the carbon exsolution from the core, according to the authors’ hypothesis, might have also contributed to the graphite crust formation in the early stage when the T at CMB was high.

Reply: Comparing the estimated carbon content in the magma ocean with the measured carbon content in the crust is not relevant to us for one major reason: the amount of graphite produced by solidification of dissolved carbon in the magma ocean should be considered as the minimum amount of graphite in

the crust because any carbon/graphite delivered to Mercury by building blocks above the saturation value for the planet would remain undissolved at the surface of the planet to form the primordial crust (possible reprocessed in the secondary volcanic crust).

This was explained in the manuscript by this sentence “*However carbon solubility in the magma ocean does fix the lower limit for the thickness of the primordial graphite crust as graphite delivered to Mercury by the building blocks may have accumulated at the surface of the molten planet at carbon-saturation⁵.*”

Moreover, the idea that “*we have a relatively good knowledge of the abundance of carbon in the crust*” is not that obvious to us when considering at the range of values proposed by different authors (<1 to > 3 wt.%). It is also unknown whether these values are laterally and vertically heterogeneous. Calculating a bulk C content for the crust seems impossible.

Xu R, Xiao Z, Wang Y, Cui J (2024) Less than one weight percent of graphite on the surface of Mercury. *Nature Astronomy* doi:10.1038/s41550-023-02169-5 (this recent reference has been added to the manuscript)

Peplowski, P. N. *et al.* Constraints on the abundance of carbon in near-surface materials on Mercury: Results from the MESSENGER Gamma-Ray Spectrometer. *Planetary and Space Science* **108**, 98-107, doi:https://doi.org/10.1016/j.pss.2015.01.008 (2015).

Peplowski, P. N. *et al.* Remote sensing evidence for an ancient carbon-bearing crust on Mercury. *Nature Geosci* **9**, 273-276, doi:10.1038/ngeo2669 (2016).

We added the following sentences in the main text: ‘Carbon solubility in reduced magma ocean (< IW-3) is extremely low when considering CO and CO₂ solubility (< 3ppm; Supplementary Fig. S8). Recent experiments have shown that it could be slightly higher (up to 15 ppm) under very reduced conditions compared to thermodynamic models¹⁹. This may be due to a minor CH₄ species in the melt.’

Some minor issues:

(1) *Quite a few figures and tables are cited incorrectly in the main text. The authors should check the order of their figures/tables, and make sure they are mentioned correctly in the appropriate position in the text.*

For example,

Line 93: Table S1 does not provide any info on the oxygen fugacity

Reply: Calculated fO₂ were reported in column E of Table S1.

Line 136: I think Fig. S8 here should be Fig. S9

Reply: Correct.

Line 139: Fig. S7 should be Fig. S8

Reply: Correct.

Line 147: Fig. S5 should be replaced by Fig. S6

Reply: Correct.

Line 155: I don't know which figure the authors refer to (Fig. S7?), but there's no Fig. 2a in the manuscript.

Reply: Correct. Reference has been made to Fig. S7

Line 305: No Table S2 (Appendix) provided

Reply: Correct. The previous sentence was modified to “The average compositions and standard deviations of analyzed phases of all experiments are shown in Table S1 (Appendix).”

Line 321 and 330: Table 2 should be Table S2 in the Supplementary Materials

Reply: Correct.

Line 452: Is the Supplementary Table S1 refer to the attached excel file which the authors sometimes refer as Appendix? Please unify the naming of all supporting figures and tables in a consistent way to avoid confusion.

Reply: Yes, the naming has been unified.

(2) Figure S10, the legend in the figure and the description in the caption do not match

Reply: Correct. The legend in the figure has been modified.

(3) Lines 127-130: I think a figure is needed for this description

Reply: We have not been able to identify any meaningful way to illustrate those probabilities. Also because those probabilities are very low, we believe that reporting the numbers gives a clearer view that most models support graphite precipitation during magma ocean solidification.

(4) Line 196: I think a word is missing here. It should be “reduced conditions”

Reply: Added as suggested, Thanks.

(5) Table S2. This table reports sulfur-free magma ocean compositions used in the thermodynamic modeling, and Table S1 also reports sulfur-free compositions used in the experiments (the added amount of FeS is not given!). So I suppose the composition for Mer8 in Table S1 should be similar to that of BSM-SiCore8 in Table S2. I think it might be better to merge the two tables into one and use the same unit for the composition or report both mol% and wt% to facilitate comparison.

Reply: A single Table S1 has been prepared. All data are expressed as wt%.

(6) Line 380: Please check the equation to make sure it's correct

Reply: The equation has been corrected for the typo (0 after alpha0)

(7) The excel file Table S1 Summary of experimental results. I think the added FeS amount for each experiment should also be listed. A quick look at the table shows that the glass composition with the highest sulfur content (~16.23%, run Mer8-2) is significantly different than the glass composition in other experiments (much lower MgO content, significantly higher CaO, Na2O and K2O content), probably due to its small fraction causing difficulty in analyzing? Would this affect the interpretation of the sulfur effect on melt liquidus?

Reply: The amount of sulfur added to the starting silicate material in each experiment was fixed (20 wt% FeS). This is now reported in the description of starting compositions. Regarding Run Mer8-2, the reviewer is correct and the high sulfur content is due to a very small percentage of melt (near-solidus experiment). The composition of this low-degree partial melt is expected to be different from high-degree partial melt, not because of difficulty in analyzing but because of different temperature, melt composition, and fO₂. More reduced conditions in that experiment are due to the high crystallization degree (phase with high SiO₂) and the relative enrichment with Si (more reduced) in the residual melt. This results in higher S solubility in the melt.

Reviewer #3 (Comments for the Author):

Summary:

The manuscript Xu et al. presents novel high-temperature and high-pressure experiments and liquidus thermodynamic modelling to understand the stability of graphite vs. diamond, finding diamond was possibly, though statistically unlikely, to be stable in Mercury's silicate magma ocean, but would be predicted to form a thick layer at the current core-mantle-boundary (CMB). A conductive diamond layer on top of the CMB would have significant implications for Mercury's thermochemical evolution given its opposite properties to an unlikely insulating FeS layer. The manuscript is extremely well written and thorough in its incorporation of literature data and plentiful supplementary figures. I recommend acceptance.

Reply: We thank the reviewer for summarizing our manuscript as a novel study and also appreciate for recommending acceptance of our paper.

Quantifying the effect of S on melting T in itself is a super interesting result worth highlighting and further study (either adding to the main text or in another manuscript rather than hidden in the supplementary). Please expand on the form of equation 12 to define the liquidus depression per wt% S. It is interesting that it is on the same order of magnitude of the effect of H₂O on olivine or plagioclase liquidi in basalt. Would be interesting to comment on whether the S effect changes with pressure too.

Reply: The form of equation 12 is non-linear (exponential) so that expressing the liquidus depression per wt% S is inappropriate. As mentioned previously, the effect of pressure on sulfur solubility was discussed in Namur et al. (2016) who did not identify any significant effect of pressure on sulfur solubility as compared to oxygen fugacity, temperature, and melt composition. Understanding precisely the (potentially minor) effect of pressure on sulfur solubility would request a specific experimental setup over a larger range of pressure conditions, which is beyond the scope of the present study. The manuscript has been modified to report that study on the effect of pressure.

We added the following sentence in the manuscript: 'It has been shown previously that sulfur solubility in reduced silicate melts is not pressure sensitive¹⁰ so that the liquidus depression we calculate as a function of sulfur content is independent on pressure conditions.'

Reviewer #4 (Comments for the Author):

In this manuscript, Xu et al. explore the possibility of a diamond-bearing layer at the core-mantle boundary in Mercury. They utilized experimental techniques to understand the geochemistry and phase stability of Mercury relevant compositions, paring the results with geophysical models to estimate the dynamics of diamond growth and mobility in the mantle and core. This is a novel idea and has the potential to have an impact on Mercury science. However, there are several major points that I believe need to be addressed before this can be suitable for publication.

Reply: We thank reviewer #4 for summarizing our manuscript as a novel idea with important impact on Mercury science. We also appreciate the recommendation to publish our paper after revision. Below we provide detailed replies to all other points the reviewer raises.

Other work has suggested that there would be an FeS layer at the core-mantle boundary (CMB) (e.g., Harder & Schubert, 2001; Smith et al., 2012; Hauck et al., 2013) as a way to explain the moment of inertia (MOI) and the abundance of S on the surface. This seems to be taking a similar approach, just from the perspective of C. Is this an alternative to the FeS hypothesis? Do these analyses fit with the MOI measurements better than a possible FeS layer? I think evidence as to how this is a more plausible hypothesis (if this is indeed the case) will strengthen this work.

[<https://doi.org/10.1006/icar.2001.6586> <https://doi.org/10.1126/science.1218809> <https://doi.org/10.1002/jgre.20091>]

Reply: The hypothesis of the FeS layer was discussed at the end of the manuscript. References to these two papers have been added:

Harder, H. & Schubert, G. Sulfur in Mercury's Core? *Icarus* 151, 118-122, (2001) doi:<https://doi.org/10.1006/icar.2001.6586>.

Hauck, S. A. et al. The curious case of Mercury's internal structure. *Journal of Geophysical Research: Planets* 118, 1204-1220, doi:10.1002/jgre.20091 (2013).

The diamond/graphite layer at CMB is not an alternative to the FeS layer. Both layers could actually coexist. In this study and in the previous work on the FeS layer (Namur et al., 2016; Cartier et al. 2020), the nature of the CMB was investigated based on phase equilibria in silicate and metallic systems, and geochemical evidence (surface composition). We believe that the thin diamond layer that we calculate would have no significant effect on the MOI. This is because the density of diamond is close to that of the lower mantle and the range of MOI values estimated for Mercury can accommodate a significant variability in mantle thickness (see Hauck et al., 2013 and Goossens et al., 2022).

Cartier, C. et al. No FeS layer in Mercury? Evidence from Ti/Al measured by MESSENGER. *Earth and Planetary Science Letters* 534, 116108, doi:<https://doi.org/10.1016/j.epsl.2020.116108> (2020).

We added the following section to the main text: “Based on geodetic data, it has been suggested that a 100 km-thick FeS layer at Mercury’s present day CMB may also exist⁶³. The physical state (solid vs. liquid) of such a layer is unknown and experimental studies combined with geochemical data have shown that this layer, if present, should be much thinner than initially suggested from geophysics^{10,64,65}. The occurrence of a diamond layer at the CMB is compatible with the presence of a FeS layer as both relate to the saturation of an element (C and S) during planetary differentiation. We propose that if the FeS layer is in a liquid state, the diamond layer would lie at the interface between the FeS layer and the silicate mantle. If the FeS layer is solid, the diamond layer would likely lie at the interface between the liquid outer metallic core and the solid FeS layer. Note however that given the strong density contrast between diamond (3500 kg/m³ and solid FeS 4840 kg/m³ in standard conditions) and the likely low viscosity of a hot, solid, FeS layer, gravitational instability may lead to overturn between the FeS layer and the diamond layer.”

-Please provide a pressure scale on Figure 3.

Reply: We would find confusing to report a pressure scale while a range of pressures at the CMB is actually possible depending on many parameters reported in Goossens et al. (2002), including the value of the MOI. Also regarding the boundary between the solid inner core and the molten outer core, Fig.3b illustrates a snapshot of a continuous process from fully molten at t_0 (4.5 Gyr) to a range of possible ICB today (radius inner core / radius whole core = 0.445 ± 0.181 ; Goossens et al., 2022). Consequently, we prefer not to follow this recommendation.

Regarding lines 152-154, are the diamonds growing and mobilizing from what is interpreted to be the liquid outer core or are diamonds co-crystallizing as the core solidifies? Clarification would be helpful. Related to this point, at what timescale would this occur at and when would a diamond layer be expected to be present at the CMB?

Reply: We are not sure to fully understand this comment. Yes, the diamonds are growing in the liquid outer core AND are crystallizing continuously as the core solidifies (because of carbon saturation in the molten metal; i.e. core solidification by crystallization of C poor phases enrich the residual liquid in C which becomes diamond saturated). Regarding the timescale, diamond would be formed as soon as the core starts crystallizing and will continue until the core is completely solid. In this paper, we discuss the graphite/diamond transition as a function of temperature at CMB. We have no way to link core crystallization to time, except by using thermochemical models (e.g. Tosi et al., 2013; Knibbe & van Westrenen, 2018). However, these thermochemical models also give a range of potential paths for temperature vs. time so that a large range of solutions may be obtained.

Tosi, N., Grott, M., Plesa, A. C. & Breuer, D. Thermochemical evolution of Mercury's interior. *Journal of Geophysical Research: Planets* 118, 2474-2487, doi:10.1002/jgre.20168 (2013).

Knibbe, J. S., & van Westrenen, W. (2018). The thermal evolution of Mercury's Fe–Si core. *Earth and Planetary Science Letters*, 482, 147-159.

To make this section clearer, we added the following section to the manuscript: ‘The formation of the solid inner core would have caused diamond to crystallize from the residual metallic molten core. This is because crystallization of C-poor phases⁵² in the solid core would have for effect to enrich the liquid outer core in C. However, the whole core being at C-saturation implies that the C-enrichment will be accommodated by forming diamond in thermodynamic equilibrium with the liquid core which would then have floated to the CMB due to the much lower density of diamond compared to the liquid iron (+Si, +C, +S) alloy.’

In the section starting at Line 70, the CMB pressure is calculated to be 5.77 GPa and in lines 108-109 it is stated that the pressure at the bottom of Mercury's magma ocean was similar to present CMB, but the experiments are run at 7 GPa. What is the rationale for this- why weren't experiments conducted at a pressure that is more similar to the predicted CMB pressure calculated in this work?

Reply: Here we refer to the answer we provided to comment (2) made by reviewer #2.

7 GPa is actually not higher than the estimated pressure range at CMB but corresponds to the maximum value of potential pressures predicted by models of Goossens et al. (2022) using the normalized polar moment of inertia of Genova et al. (2019). It is true that pressure tends to favor the stabilization of

diamond at constant temperature. But a higher pressure also implies a higher liquidus temperature which does not favor diamond stability (but favors graphite stability).

Running experiments at 7 GPa enabled us to test the most favorable conditions to produce diamond. This does not imply that our conclusions are forced to support diamond stability because our experimental results into models to interpolate the diamond-graphite stability for the whole range of potential pressures predicted by models of Goossens et al. (2022).

As shown in Figure 1a of the paper, the temperature of the diamond-graphite transition is also close to the liquidus of the two starting compositions (Mer8 and Mer15) and allowed us to calibrate the pressure of our experimental setup (by identifying a graphite to diamond transition of the graphite capsule). Additionally, a suite of experiments on sulfur solubility under reduced conditions were already available at pressure <4GPa, enabling us to interpolate any results between our experimental pressure (7GPa) and those of Namur et al. (2016).

-At Mercurian CMB pressure conditions and a range of relevant temperature (early and present) this is sitting at metastability for diamond and graphite which is mentioned in line 176. I am not convinced at the possible longevity of this layer, lasting long enough to be measurable via MOI today. If some of the layer were to transform to graphite, it would be gravitationally unstable and reduce the thickness of the layer, thus dampening the effect of this diamond bearing layer. Additionally, convection in the first Ga of Mercury's history could disrupt a diamond layer and redistribute in the mantle.

Reply: We thank the reviewer for pointing out this interesting idea. We agree with reviewer #4 that MOI measurements may not be able to identify the occurrence of such a layer at CMB. Actually we believe that a diamond layer would have no effect on the MOI. Here we use constraints from phase equilibria and surface composition (carbon saturation of the planet) to discuss the speciation of carbon at CMB, i.e. graphite vs. diamond.

We changed as follows: ‘Future work should investigate how the presence of a km-scale diamond layer at the CMB would affect the thermal evolution of the silicate fraction of Mercury⁶⁰. We however believe that such a thin layer could not be unambiguously confirmed using interior models fitting Mercury’s MOI and tidal Love numbers³⁰ given the current large uncertainties on the physical properties of Mercury’s mantle and core^{24,25,30}’.

Gravitational instability of an early graphite layer accumulated at CMB (before transition to diamond) is indeed an interesting working hypothesis that will be inspired by the work we propose. But again, this question is beyond the scope of this study. In addition, transformation of diamond deep in the mantle to a thinner graphite layer at the top of the mantle would have a non-measurable effect on the MOI.

We added the following section: ‘The potential diamond layer thicknesses shown in Fig. 2 are thus maximum values because the extent of the graphite to diamond reaction cannot be predicted and because some early formed graphite may have been redistributed in the mantle due to strong convection and density contrast between graphite and silicate’.

The last part of the comment “Additionally, convection in the first Ga of Mercury’s history could disrupt a diamond layer and redistribute in the mantle.” is actually discussed in lines 213-215 of the paper.

We added the following section to the manuscript: In addition, convection in the lower mantle may have disrupted the diamond layer and redistributed diamond in the mantle or even in the crust. However, we believe that graphite and diamond redistribution in the silicate part of the planet was likely minimal. This is because strong convection in the lower mantle stopped at 3.7 Ga⁴⁵ and that inner crystallization

only started at about 4 Ga⁶². Most of the diamond layer or its graphite precursor therefore formed after strong mantle convection stopped which limits the extent to which the diamond layer may have been disrupted and the extent to which graphite may have been redistributed by gravitational instability.

The density of the solidified magma ocean and diamonds needs to be scaled with pressure. With the minerals being considered (e.g., opx, cpx, garnet, olivine) the density is likely greater than the 3000 kg/m³ being considered when solidified. This could be an issue thinking about gravitational stability of diamond below a potentially denser silicate at the CMB after the magma ocean stage.”

Reply: We agree that the density of the silicate mantle should be in the range 3000-3200 kg/m³. Indeed, the mantle of Mercury is made up forsterite (density at 5.5 GPa and 1700 K is 3200 kg/m³), enstatite (density at 5.5 GPa and 1700 K is 3243 kg/m³) and sulfides (density ca. 2800 kg/m³). The density of diamond at the CMB is at 5.5 GPa and 1700 K is 3490 kg/m³. There is thus no reason to expect any gravitational instability between a dense lower layer of diamond and a less dense upper layer made of silicate minerals. It seems from this comment that the reviewer believes diamond is a light phase.

We added the following sentence: ‘Diamond is not expected to be affected by gravitational instability because this phase is denser than forsterite and enstatite, the main silicate minerals of the present day mantle.’

Figure S2a- it looks like there is a mid-gray phase surrounding the large phase that is labeled “Sf” (and some of the smaller sulfides). In experiments that are rich in Fe and S that are at lower temperatures (which S2a is the lowest T) there is phase separation of a sulfide-rich rim with a metal (Fe)-rich core, evolving to a single phase with increasing temperature. Is this the case for this experiment that there are indeed two phases? If so, it would be beneficial to label the two phases appropriately.

Reply: We thank the reviewer for pointing this out. The mid-gray phase surrounding the Sf phase is a (Mg,Fe,Ca)S sulfide. It is labelled in the Figure and note in the caption as well.

Lines 275-275- what assembly was used (e.g., 14/8)? Were WC cubes used for these experiments?

Reply: We apologize for this unclear description of our experimental assembly. The assembly consist of a natural pyrophyllite cubic block (the cube edge length: 38.5 mm), semi-sintered magnesium oxide sleeves, and a 10 mm inner diameter graphite furnace, which is a typical assembly for the cubic press – it is different with an oft-used multi-anvil press. Yes, the six anvils of cubic press are made of WC, which serve as WC cubes and directly touch the experimental assembly. The detailed description of this kind of press can be found in Wu et al. (2024) [A novel rapid cooling assembly design in a high-pressure cubic press apparatus, Matter Radiat. Extremes 9, 027402]. The ref. is cited now.

REVIEWERS' COMMENTS

Reviewer #2 (Remarks to the Author):

The authors did a great job in responding to the reviewers' comments. All my comments/suggestions have been satisfactorily addressed. Thus, I think the manuscript can be accepted for publication in its current form.

Reviewer #2 (Remarks on code availability):

It seems like enough information is provided for the code in the repository, but I haven't tried to run the code myself.

Reviewer #4 (Remarks to the Author):

This is my second review of the manuscript from Xu et al. on "A diamond-bearing core-mantle boundary on Mercury". The authors provided thorough responses to my comments and addressed any issues. I have no further comments, thus I recommend that this manuscript be accepted.

Response to reviewer comments

[Referee comments in *italics*, responses in normal font and blue]

Reviewer #1 (Remarks to the Author):

The authors did a great job in responding to the reviewers' comments. All my comments/suggestions have been satisfactorily addressed. Thus, I think the manuscript can be accepted for publication in its current form.

Reply: We thank that the reviewer for satisfying our responses to all of his/her comments and appreciate that reviewer #1 to recommend for accepting our paper in its current form.

Reviewer #2 (Remarks on code availability):

It seems like enough information is provided for the code in the repository, but I haven't tried to run the code myself.

Reply: We thank that the reviewer for satisfying our responses to his/her comments. We double check the code and confirm the code can run well.

Reviewer #4 (Remarks to the Author):

This is my second review of the manuscript from Xu et al. on "A diamond-bearing core-mantle boundary on Mercury". The authors provided thorough responses to my comments and addressed any issues. I have no further comments, thus I recommend that this manuscript be accepted.

Reply: We thank that the reviewer for agreeing that we provided thorough responses to all of his/her comments and addressed all issues, and thanks for recommending our manuscript for acceptance.